# Involvement of the AKT Pathway in Resistance to Erlotinib and Cabozantinib in Triple-Negative Breast Cancer Cell Lines

**DOI:** 10.3390/biomedicines11092406

**Published:** 2023-08-28

**Authors:** Cory Lefebvre, Sierra Pellizzari, Vasudeva Bhat, Kristina Jurcic, David W. Litchfield, Alison L. Allan

**Affiliations:** 1London Regional Cancer Program, London Health Sciences Centre, London, ON N6A 5W9, Canada; clefebvre2019@meds.uwo.ca (C.L.); spelliz2@uwo.ca (S.P.); vbhat@uwo.ca (V.B.); 2Department of Anatomy & Cell Biology, Western University, London, ON N6A 3K7, Canada; 3Department of Biochemistry, Western University, London, ON N6A 3K7, Canada; kjurcic@uwo.ca (K.J.); litchfi@uwo.ca (D.W.L.); 4Department of Oncology, Western University, London, ON N6A 3K7, Canada; 5Lawson Health Research Institute, London, ON N6A 5W9, Canada

**Keywords:** triple-negative breast cancer (TNBC), epidermal growth factor receptor (EGFR), hepatocyte growth factor receptor (c-Met/HGFR), AKT1, erlotinib, cabozantinib, therapeutic resistance, quantitative mass spectrometry proteomics, kinomics, phosphoproteomics

## Abstract

Resistance to protein tyrosine kinase inhibitors (TKIs) presents a significant challenge in therapeutic target development for cancers such as triple-negative breast cancer (TNBC), where conventional therapies are ineffective at combatting systemic disease. Due to increased expression, the receptor tyrosine kinases EGFR (epidermal growth factor receptor) and c-Met are potential targets for treatment. However, targeted anti-EGFR and anti-c-Met therapies have faced mixed results in clinical trials due to acquired resistance. We hypothesize that adaptive responses in regulatory kinase networks within the EGFR and c-Met signaling axes contribute to the development of acquired erlotinib and cabozantinib resistance. To test this, we developed two separate models for cabozantinib and erlotinib resistance using the MDA-MB-231 and MDA-MB-468 cell lines, respectively. We observed that erlotinib- or cabozantinib-resistant cell lines demonstrate enhanced cell proliferation, migration, invasion, and activation of EGFR or c-Met downstream signaling (respectively). Using a SILAC (Stable Isotope Labeling of Amino acids in Cell Culture)-labeled quantitative mass spectrometry proteomics approach, we assessed the effects of erlotinib or cabozantinib resistance on the phosphoproteome, proteome, and kinome. Using this integrated proteomics approach, we identified several potential kinase mediators of cabozantinib resistance and confirmed the contribution of AKT1 to erlotinib resistance in TNBC-resistant cell lines.

## 1. Introduction

Breast cancer is the most commonly diagnosed cancer among women and the second leading cause of cancer death in women [1]. Triple-negative breast cancer (TNBC) accounts for approximately 15–20% [2] of these breast cancer cases. The triple-negative classification refers to breast tumors lacking expression of the receptors ER (estrogen receptor), PR (progesterone receptor), and HER2/neu (human epidermal growth factor receptor 2) [3,4], which are commonly used as therapeutic targets for systemic treatments. For most patients with breast cancer, treatment entails surgery with radiation therapy and/or systemic therapy, which, depending on subtype, includes targeting of hormone receptor and growth factor receptor pathways with anti-estrogen/progesterone (tamoxifen, aromatase inhibitors) and/or anti-HER2 therapies (Herceptin^®^/trastuzumab, lapatinib) [5,6]. However, TNBC currently does not respond to these molecular-targeted therapies [7,8,9] and tends to have a worse prognosis compared to other breast cancer subtypes [10,11]. There is, therefore, a critical need to identify new therapeutic targets in TNBC.

The search for potential therapeutic targets in TNBC has led to the study of receptor tyrosine kinases (RTKs), such as epidermal growth factor receptor (EGFR) and hepatocyte growth factor receptor (c-Met), which are often overexpressed in TNBC tumors [12,13]. Overexpression of EGFR is associated with poor outcomes in various malignancies [14,15,16], including breast cancer [14,17], and EGFR overexpression is more frequently found in triple-negative/basal-like breast cancer compared to other molecular subtypes [18,19,20]. Similarly, c-Met has been shown to be more highly expressed in basal-like [21,22] and triple-negative breast cancers [2,12,23] relative to other breast cancer subtypes. Dysregulation of either EGFR [24,25] or c-Met [12,26,27] pathways play an important role in tumorigenesis and metastasis via their regulation of cell behaviors such as proliferation, motility, apoptotic evasion, and invasion.

Given the reliance of cancer progression on these pathways, EGFR and c-Met present as potential effective targets in TNBC. However, clinical studies evaluating the use of the anti-EGFR and anti-c-Met inhibitors in triple-negative breast cancer often demonstrate an initial response among a subset of patients followed by the development of treatment resistance [28,29]. Erlotinib (Tarceva^®^, OSI-774, R1415, CP358774, NSC718781) is a reversible small molecule ATP-competing inhibitor of EGFR primarily used to treat non-small cell lung cancer (NSCLC) [30]. By binding to the ATP-binding pocket of EGFR, erlotinib prevents trans-autophosphorylation and activation of downstream cell cycle progression, proliferation, and angiogenesis signaling pathways [31]. In the clinical trial setting, erlotinib has demonstrated initial activity in a subset of breast cancer patients from phase I and phase II clinical trials [32,33,34], but it is often followed by disease progression through treatment. Cabozantinib (XL184, BMS907351) is a small molecule ATP-competitive inhibitor of c-Met and VEGFR2 that is clinically used for the treatment of medullary thyroid cancer and hepatocellular carcinoma. In preclinical studies, cabozantinib has been shown to inhibit the phosphorylation and activation of c-Met and downstream signaling molecules in vitro [35] and in vivo [36,37], resulting in the inhibition of cell proliferation and tumor growth. Similar to EGFR inhibitors, results from phase II clinical trials in metastatic breast cancer report cabozantinib having an initial clinical activity that is often followed by disease progression [38,39]. Although erlotinib and cabozantinib may have some clinical activity in controlling disease progression at treatment onset, ultimately, breast tumors continue to progress and develop resistance to treatment. However, the mechanisms underlying this resistance remain poorly understood.

Treatment resistance can be categorized at the cellular level as intrinsic or adaptive resistance and at the tumor level as innate or acquired resistance [40]. Intrinsic resistance refers to the a priori lack of measurable response to the inhibitor, whereas acquired resistance refers to the development of treatment resistance following an initial response to the inhibitor. Resistance to kinase inhibition can be explained by a number of underlying molecular mechanism(s). A common and well-studied mechanism is the presence of mutations—either innate or acquired—in the target kinase that overcomes the activity of the kinase inhibitor. Often, these are point mutations in the ATP binding pocket known as “gatekeeper” mutations, such as the T790M mutation in erlotinib-resistant non-small cell lung cancer (NSCLC) [41,42]. Other reported mechanisms involve drug efflux pumps [43], epithelial-to-mesenchymal transition (EMT), and stem cell-like phenotypes [44].

One of the most common mechanisms of acquired resistance is the development of adaptive changes in downstream and/or redundant signaling pathways that compensate for the loss of signaling from the target kinase. This often involves the upregulation of kinases within the kinome—the collection of all kinases. In erlotinib-resistant NSCLC tumors, the upregulation of c-Met [45,46] and HER2 [47] are common resistance mechanisms alongside the T790M mutation. This upregulation may be due to genetic amplification, increased gene copy number, or increased activity of the compensatory pathway. Kinases involved in the resistance to EGFR inhibitors have been extensively studied in NSCLC and lung adenocarcinomas, revealing many kinases that contribute to a resistant phenotype, including c-Met, HER2, HER3, VEGFR, PDGFR, PI3K, AKT, FAK, JAK, and ERK [45,48]. Given the more recent development and clinical use of c-Met inhibitors in comparison, resistance mechanisms have not been as thoroughly studied, but kinases, including EGFR, HER3, PAK1, and PI3K [49,50,51], have been reported to be involved in resistance. However, resistance to inhibitors targeting EGFR and c-Met has not been well characterized in TNBC. Given that kinases often play a role in the development of resistance to tyrosine kinase inhibitors in cancer [45,46,52,53], there is an opportunity to use quantitative mass spectrometry-based kinomics and phosphoproteomics approaches to assess global changes in kinase expression and activity in erlotinib- and cabozantinib-resistant TNBC.

In the current study, we hypothesized that adaptive responses in regulatory kinase networks within the EGFR and c-Met signaling axes contribute to the development of acquired erlotinib and cabozantinib resistance in TNBC. We developed two separate models for erlotinib and cabozantinib resistance using the MDA-MB-231 and MDA-MB-468 human TNBC cell lines, respectively. The acquired resistance models were established by applying long-term exposure to and selective pressure on cultured TNBC cells using a clinically relevant concentration of cabozantinib or erlotinib (defined as a “low-resistance” model) or long-term exposure and selective pressure using a step-wise escalation method to a supra-clinical concentration (defined as a “high-resistance” model) [54]. Comparing these resistant models to sensitive parental controls, we observed that erlotinib- or cabozantinib-resistant cell lines demonstrate enhanced cell proliferation, migration, invasion, and/or activation of canonical downstream signaling. Using a SILAC (Stable Isotope Labeling of Amino acids in Cell Culture)-labeled quantitative mass spectrometry proteomics approach, we assessed the effects of erlotinib or cabozantinib resistance on the phosphoproteome, proteome, and kinome via an integrated analysis approach using KSEA and modified KSEA (KSEAM) to assess kinase activity changes. We identified an upregulation of AKT1, CSNK2A1, and ERK1 activity and SYK expression in erlotinib-resistant cells and an upregulation in CDK1, CDK7, and CK2A1 activity in cabozantinib-resistant cells. Functional inhibitor studies revealed inhibitor synergy between erlotinib and AKT inhibitor VIII in MDA-MB-468 TNBC cells, suggesting that AKT1 contributes to erlotinib resistance in our resistant MDA-MB-468 cell model and that information gained from integrated phosphoproteomic and kinomics analysis can be applied for therapeutic benefit.

## 2. Materials and Methods

### 2.1. Cell Culture and Breast Cancer Therapy Resistance Models

The human TNBC breast cancer cell lines MDA-MB-468 and MDA-MB-231 were obtained from Dr. Ann Chambers (London Health Sciences Centre, London, ON, Canada) and were cultured under standard conditions in 5% CO_2_ at 37 °C in Dulbecco’s modified Eagle’s medium (DMEM) and nutrient mixture F12 (1:1) (ThermoFisher, Waltham, MA, USA) with 10% fetal bovine serum (FBS; VWR Life Science, Radnor, PA, USA), and minimum essential medium Eagle-alpha modifications (α-MEM; ThermoFisher) with 10% FBS, respectively. To develop resistant cell line models (Appendix A), MDA-MB-468 cells were subjected to long-term culture in the presence of the anti-EGFR inhibitor erlotinib (LC Laboratories, Wobourn, MA, USA), and MDA-MB-231 cells were cultured with the anti-c-Met inhibitor cabozantinib (MedChemExpress, Monmouth Junction, NJ, USA) for at least 10–12 sub-passages. For the low-resistance (LR) model, cells were cultured with a clinically relevant concentration (2.9 μM erlotinib, 3.9 μM cabozantinib) for 10–12 sub-passages. For the high-resistance (HR) model, concentrations of inhibitors were increased in a step-wise fashion at least every 3 sub-passages starting at 1 μM, followed by 5 μM, then 15 μM. A sensitive (S) control model was developed in parallel to the resistant cell lines by culturing the same cell lines with an equivalent volume of dimethyl sulfoxide (DMSO; Sigma-Aldrich, Mississauga, ON, Canada). Resulting cell models were named 468LR, 468HR, 231LR, or 231HR (resistant) and 468S or 231S (sensitive). When assessing EGFR and c-Met signaling, cells were incubated with 100 ng/mL EGF (Sigma-Aldrich, Mississauga, ON, Canada) or 50 ng/mL HGF (Abcam, Cambridge, UK), respectively.

### 2.2. Colony-Forming Assays

Colony-forming assays were performed to assess the development of erlotinib and cabozantinib resistance. Cells were seeded in 6-well plates at a density of 100–150 cells/well, allowed to attach for several hours, and incubated with either inhibitor (0.1 nM–100 μM erlotinib; 1 nM–100 μM cabozantinib) or DMSO. Colonies were allowed to develop over 10–14 days at 37 °C, 5% CO_2_, with media containing inhibitors/DMSO refreshed every 4 days. At the assay endpoint, colonies were fixed and stained with 6% glutaraldehyde + 0.5% crystal violet solution. The mean number of colonies (defined as >50 cells) per well was calculated using FIJI ImageJ software (NIH, version 1.52p, Bethesda, MD, USA; RRID:SCR_002285) and used to calculate the surviving fraction. Using a non-linear regression, the IC50 values were determined for each cell line-inhibitor combination, and cell lines were considered resistant if there was at least a 1.5-fold increase in the resistance factor (ratio of resistant IC50 to sensitive IC50).

### 2.3. Cell Migration and Invasion Assays

Differences in cell migration and invasion between resistant and sensitive cell models were assessed using transwell migration assays. Falcon^®^ transwell inserts (8 µm pore size) were placed in 24-well dishes, coated with gelatin (migration) or Matrigel™ (invasion; 15% *v*/*v*; BD Science, San Jose, CA, USA) and exposed to culture media (10% FBS) in the bottom well. Cells (5 × 10^4^ cells/well) were seeded onto the top portion of each transwell chamber and treated with inhibitors (5 µM erlotinib or 5 µM cabozantinib), vehicle control (DMSO), or proliferation control (mitomycin C). Following a 24-h incubation at 37 °C and 5% CO_2_ under the different treatment conditions, transwells were fixed with 1% glutaraldehyde and mounted and stained with Vectashield^®^ Antifade Mounting Medium with DAPI (Vector Laboratories). Five high-powered fields of view (HP-FOVs) were analyzed for each well, and the mean number of migrated or invaded cells per FOV was calculated using FIJI ImageJ software.

### 2.4. Immunoblotting

Cells were treated with inhibitors (5 µM erlotinib or 5 µM cabozantinib) or vehicle control (DMSO) for 24 h prior to harvesting of cell lysates for immunoblotting. This time point was chosen based on previously reported time-course studies in our laboratory [55]. The protein concentration of cell lysates was quantified by a Lowry assay, and 30 μg/sample was boiled for 10 min with sodium dodecyl sulfate (SDS), subjected to SDS polyacrylamide gel electrophoresis (SDS-PAGE) at 150 V for 1 h, and transferred onto polyvinylidene difluoride (PVDF) membranes (Millipore, Sigma-Aldrich). Membranes were blocked using 5% bovine serum albumin (BSA) in Tris-buffered saline (TBST) + 0.1% Tween-20. A summary of the primary antibodies used can be found in Appendix A. Secondary antibodies included goat anti-mouse IgG, or goat anti-rabbit IgG secondary antibodies (Calbiochem, Billerica, MA, USA) conjugated to horseradish peroxidase were used at a concentration of 1:1000, diluted in 5% BSA in TBST. Protein expression was visualized using Amersham^TM^ ECLTM Prime Detection Reagent (GE Healthcare Lifesciences, Wauwatosa, WI, USA).

### 2.5. SILAC Media Formulation and Incorporation

SILAC-dropout Dulbecco’s minimum essential medium (DMEM; Cat: 319-119-CL, Wisent, QC, Canada) lacking L-arginine and L-lysine was supplemented with isotope-encoded L-arginine (13C6) and L-lysine (4,4,5,5-D4) (CLM-2265 and DLM-2640, respectively (Cambridge Isotope Laboratories Inc., Tewksbury, MA, USA)) at respective concentrations of 86.2 mg/L (0.398 mM) and 61.16 mg/L (0.274 mM) to create a “heavy-label” medium. For the “light-label” medium, SILAC-dropout DMEM was supplemented with equivalent molar amounts of unlabeled L-arginine (83.9 mg/L) (A4599; Sigma-Aldrich, Saint Louis, MO, USA) and L-lysine (60.04 mg/L) (L7039; Sigma-Aldrich). Both “heavy-label” and “light-label” SILAC media were supplemented with 10% (*v*/*v*) 10 kDa-dialyzed FBS (Wisent), penicillin (100 U/mL), (Life Technologies, Carlsbad, CA, USA), and L-proline (400 mg/L) (P8865, Sigma-Aldrich) in order to prevent arginine to proline conversion [56,57]. Media was filter-sterilized prior to use for cell culture. The incorporation of SILAC labels in resistant and sensitive cells was adapted over 10 sub-passages, and SILAC incorporation efficiency was assessed by mass spectrometry. SILAC-labeled MDA-MB-468 models were treated with 5 μM erlotinib, and SILAC-labeled MDA-MB-231 cell models were treated with 5 μM cabozantinib for 24 h prior to the collection of protein lysates for phosphoproteomic, proteomic, and kinomic sample preparation. This included 3 biological replicates with “light”-control and “heavy”-erlotinib conditions and 3 biological replicates with labels swapped.

### 2.6. LC-MS/MS Data Acquisition and Processing

SILAC-labeled cells were treated with inhibitors (5 µM erlotinib or 5 µM cabozantinib) or vehicle control (DMSO) for 24 h prior to harvesting for LC-MS/MS analysis. The LC-MS/MS workflow and integrated analysis approach are summarized in Appendix A. For data-dependent acquisition (DDA) LC-MS/MS, phosphoproteomic and proteomic digested peptides were analyzed using a nano-HPLC (High-performance liquid chromatography) coupled to an Orbitrap Fusion™Lumos™Tribid™ mass spectrometer. Sample amounts of 0.9 ug were used for phosphoproteome acquisition and 0.5 μg was used for whole proteome acquisition. Coated nano-spray emitters were generated from fused silica capillary tubing, with 75 µm internal diameter, 365 µm outer diameter, and 5–8 µm tip opening, using a laser puller (Sutter Instrument Co., Novato, CA, USA, model P-2000, with parameters set as heat: 280, FIL = 0, VEL = 18, DEL = 2000). Nano-spray emitters were packed with C18 reversed-phase material (Reprosil-Pur 120 C18-AQ, 1.9 µm) and resuspended in methanol using a pressure injection cell. The sample in 5% formic acid was directly loaded at 400 nL/min for 20 min onto a 75 µm × 15 cm nano-spray emitter. Peptides were eluted from the column with an acetonitrile gradient generated by an Eksigent ekspert™ nanoLC 425 and analyzed on an Orbitrap Fusion™Lumos™Tribrid™. The total DDA protocol was 240 min for phosphoproteome and 300 min for whole proteome acquisition. The MS1 scan had an accumulation time of 50 ms within a mass range of 400 to 1500 Da, using an orbitrap resolution of 120,000, 60% RF lens, AGC target of 125%, and 2400 volts. This was followed by MS/MS scans with a total cycle time of 3 s. An accumulation time of 50 ms and 33% HCD collision energy was used for each MS/MS scan. Each candidate ion was required to have a charge state from 2 to 7 and an AGC target of 400%, isolated using an orbitrap resolution of 15,000. Previously analyzed candidate ions were dynamically excluded for 9 s. Kinomic peptide samples were loaded onto a nano-HPLC (high-performance liquid chromatography) system coupled to an Orbitrap Fusion™Lumos™Tribid™ mass spectrometer for data-dependent acquisition (DDA) LC-MS/MS. Mass spectrometric data were analyzed using MaxQuant (v1.6.2.10, Max Planck Institute of Biochemistry, Munich, Germany) [58,59] and searched against the Human Swissprot-Uniprot FASTA database (homo sapiens), and MaxQuant output files were interpreted using Perseus (v1.6.15.0, Max Planck Institute of Biochemistry, Munich) [60]. Detailed LC-MS/MS data acquisition and processing methodology is provided in the Appendix A.

### 2.7. Bioinformatics–Kinase Activity Analysis

To assess changes in kinase activity, KSEA (Kinase-Substrate Enrichment Analysis) [61] and KSEAM (modified KSEA) techniques were used. The groups of significantly increasing phosphosites and significantly decreasing phosphosites were input separately into the KEA2 web portal (https://www.maayanlab.net/KEA2/, accessed on 10 August 2021), which mines literature kinase-substrate databases for upstream kinases of input phosphosites and calculates an enrichment score for associated kinases. For KSEA, all quantified phosphosites were inputted into the KSEA web app portal [62], querying PhosphositePlus and NetworKIN (minimum score cut-off = 2) and FDR of 0.1 to generate z-scores of the enrichment. The modified KSEA analysis works on the same basis as KSEA using PhosphositePlus and NetworKIN predictive databases to associate substrates to upstream kinases but also inputs the change in kinase expression measured from kinomics or proteomic datasets. The scores are calculated using the following formula:$$KSEAM\;score=\frac{(r-\bar{r})\surd m}{\delta_{r}},\;where\;r=\frac{\bar{s}}{k}$$
where *r* represents the kinase activity ratio of the mean phosphorylation fold change ($\bar{s}$) to the expression fold change (k) of the respective upstream kinase, where $\bar{r}$ is the average calculated ratio for all identified kinases, where *m* represents the number of phosphosite substrates identified per kinase, and $\delta_{r}$ represents the standard deviation of average kinase activity ratio.

### 2.8. Drug Combination Synergy Assays

To validate a strategy for inhibitor combination synergy suggested by the integrated proteomics analysis, erlotinib-resistant (468LR, 468HR) and sensitive (468S) cells were treated with erlotinib combined with AKT inhibitor VIII (Sigma-Aldrich, St. Louis, MO, USA) or the ERK1/ERK2 inhibitor Temuterkib (MedChemExpress, Monmouth Junction, NJ, USA). Colony-forming assays were performed as described above, except each dose of erlotinib (0.1 nM–1 μM) was incubated in combination with each dose (0.1 nM–1 μM) of AKT inhibitor VII or Temuterkib. Drug combination synergy scores–HSA (Highest Single Agent), ZIP (Zero Interaction Potency), Loewe, Bliss scores, and drug combination sensitivity scores were calculated using the SynergyFinder web application [63,64].

### 2.9. Statistical Analysis

All in vitro experiments were performed with a minimum of 3 biological replicates, with at least 3 technical replicates included in each experiment. For analysis of mass spectrometry experiments, please refer to Section 2.7 and Section 2.8. Quantitative data were compiled from all experiments. Unless otherwise noted, data are presented as the mean ± SD. Statistical analysis was performed using GraphPad Prism 8.0 software (GraphPad Software, San Diego, CA, USA) using one-way or two-way analysis of variance (ANOVA) with Tukey post-tests (for comparison among all treated conditions) and non-linear regression (for IC50 dose-curves). Values of *p* ≤ 0.05 were considered statistically significant.

## 3. Results

### 3.1. Erlotinib-Resistant Human MDA-MB-468 TNBC Cells Demonstrate a Loss of Inhibition of Cell Proliferation, Migration, Invasion, and EGFR/ERK/AKT Signaling

Based on our previous work [55], we chose the MDA-MB-468 human TNBC cell line as a drug-sensitive parental cell line with which to develop the erlotinib-resistant cell models. The clinically relevant dose for the low-resistance (468LR) model was determined from the maximum patient plasma concentration of erlotinib (1.14 μg/mL or 2.89 nmol/mL) determined in human Phase I single-dose safety trials [65]. Using colony-forming assays, we observed a significant increase in the IC50 values of 468LR (IC50 = 1.84 μM, R^2^ = 0.89) and 468HR (IC50 = 1.18 μM, R^2^ = 0.78) cell models compared to 468S cells (IC50 = 0.13 μM, R^2^ = 0.83) in response to erlotinib (Figure 1A). This corresponds with resistance factors (resistant IC50/sensitive IC50) of 13.7× (468LR) and 8.8× (468HR), which cross the arbitrary threshold of 1.5× to designate resistance [54]. In addition to cell growth effects, we next wanted to assess if the resistant models also acquired resistance to the anti-migratory and anti-invasive effects of erlotinib. We observed that although migration of 468S and 468LR cells was still significantly attenuated in response to erlotinib relative to control (*p* ≤ 0.05), erlotinib had no such inhibitory effect on the 468HR cells (Figure 1B). Invasion of both 468LR and 468HR models was also not inhibited by erlotinib, whereas 468S cells demonstrated significant attenuation of invasion relative to control in response to erlotinib (*p* ≤ 0.05) (Figure 1C). To assess whether the introduction of resistance-modulated erlotinib influences EGFR signaling, we assessed the expression of phosphorylated and total EGFR protein and its downstream effectors ERK1/2 and AKT1 through immunoblotting. We observed a significant decrease in EGFR and ERK1/2 phosphorylation in 468S cells relative to control in response to erlotinib (*p* ≤ 0.05), whereas no response/change was observed in either 468LR or 468HR cells (Figure 1D; Appendix A). Interestingly, we observed a significant increase in AKT1 phosphorylation (Ser473 and Thr308) in erlotinib-treated 468HR cells relative to 468S and 468LR cells (*p* ≤ 0.05) (Figure 1D; Appendix A). Parallel to this, we also observed a significant increase in c-MET protein expression (one of the classical activation mechanisms of the AKT pathway [66]) in the resistant 468LR and 468HR cells compared to the sensitive 468S cells (Appendix A). Taken together, this suggests a potential acquired compensatory mechanism underlying the observed resistance to erlotinib in MDA-MB-468 cells.

### 3.2. Cabozantinib-Resistant Human MDA-MB-431 TNBC Cells Demonstrate a Lack of Inhibition of Cell Proliferation, Invasion, and AKT Signaling

Based on our previous work [55], we chose the MDA-MB-231 human TNBC cell line as a drug-sensitive parental cell line with which to develop the cabozantinib-resistant cell models. The clinically relevant dose for the low-resistance model was determined from the maximum plasma concentration of cabozantinib (1.93 μg/mL or 3.89 nmol/mL) determined in Phase 1 single-dose safety trials [67]. Using colony-forming assays, we observed an increase in the IC50 values of the 231LR (IC50 = 3.44 μM, R^2^ = 0.87) and 231HR (IC50 = 2.51 μM, R^2^ = 0.92) cell models compared to 231S cells (IC50 = 1.33 μM, R^2^ = 0.91) in response to cabozantinib (Figure 2A). This corresponds with resistance factors of 2.6× and 1.9× for the 231LR and 231HR models (respectively), which again crosses the arbitrary threshold of 1.5× to designate resistance. Although migration of all three models (231S, 231LR, 231HR) remained sensitive to cabozantinib inhibition relative to control (*p* ≤ 0.05) (Figure 2B), invasion of 231HR cells was not inhibited by cabozantinib (Figure 2C). To assess whether the introduction of resistance modulated the effects of cabozantinib on c-Met signaling, we assessed the expression of phosphorylated and total protein of c-Met and its downstream effectors ERK1/2 and AKT1 through immunoblotting. Not unexpectedly (based on previous findings) [55], we did not observe any significant differences in c-Met or ERK1/2 phosphorylation in response to cabozantinib treatment (Figure 2D; Appendix A). However, there was a significant attenuation in AKT1 phosphorylation of Ser473 but not Thr308 in cabozantinib-treated 231S cells (*p* ≤ 0.05) that was not present in the cabozantinib-resistant cell lines (231LR and 231HR) (Figure 2D; Appendix A).

### 3.3. Acquisition of Resistance to Erlotinib and Cabozantinib Does Not Alter the Breast Cancer Stem Cell Phenotype

Resistance to cancer therapeutics (including tyrosine kinase inhibitors) can be driven by a variety of potential mechanisms, including increased activity of drug efflux pumps, increased cell plasticity (i.e., cancer stem cell [CSC] or epithelial-to-mesenchymal transition [EMT] phenotypes), and/or adaptive changes to signaling networks [43,44]. To examine the mechanisms underlying the acquired resistance to erlotinib and cabozantinib in our TNBC models, we first assessed drug efflux pump activity (ABCB1/MDR1, ABCC1/MRP1, and ABCG2/BCRP; Appendix A) and CSC/EMT phenotype (Appendix A). We did not observe any significant differences in drug pump activity or CSC/EMT phenotypes between sensitive versus resistant TNBC cell models, ruling these out as potential mechanisms driving the observed RTK resistance.

### 3.4. Acquisition of Resistance to Erlotinib and Cabozantinib Alters the Proteome and Kinome of TNBC Cells with Minimal Change to the Phosphoproteome

We next employed an integrated phosphoproteomic and kinomics workflow (Appendix A) to investigate whether changes in kinase expression and/or activity were involved in the development of resistance to erlotinib (Figure 3A–E) or cabozantinib (Figure 4A–E), specifically in the low-resistance TNBC models (468LR and 231LR) as compared to respective sensitive models (468S and 231S). Due to technical difficulties in incorporating SILAC labels in the HR models, we opted to compare the LR-resistant models to the respective sensitive cells. Cells were SILAC-labeled with an efficiency of >92% (Appendix A) and subjected to LC-MS/MS analysis for investigation of the proteome, phosphoproteome, and kinome, as described in Materials and Methods. Comparative analysis of 468S and 468LR TNBC cells treated with erlotinib (5 µM) revealed 2159 protein groups quantified from the proteomics, 879 protein groups from the kinomics, and 555 protein groups with quantified phosphorylation sites (*n* = 1023) from the phosphoproteomics (Figure 3A). Comparative analysis of 231S and 231LR TNBC cells treated with cabozantinib (5 µM) revealed 2189 protein groups quantified from the proteomics, 724 protein groups from the kinomics, and 223 protein groups with quantified phosphorylation sites (*n* = 404) from the phosphoproteomics (Figure 4A). There was considerable overlap of proteins quantified across the six replicates in both kinomics and proteomic datasets within each experiment comparing the resistant to sensitive cell lines. When assessing the distribution of quantified proteins from the erlotinib-resistant kinomics/proteomics (Figure 3B,C) and cabozantinib-resistant kinomics/proteomics (Figure 4B,C), the majority of proteins fell within the bounds of log2 mean fold-change of −0.59 and 0.59 (threshold for biological significant fold-change), indicating that the observed acquisition of RTK resistance results in select proteins that show significant alteration in expression. In contrast, analysis of the resistant versus sensitive cell models indicates minimal disruption in the phosphoproteome as a result of the acquisition of erlotinib (Figure 3D) or cabozantinib (Figure 4D) resistance. The average fold-change expression of phosphorylation sites was plotted against the corresponding protein ratio to confirm that the change in the phosphoproteome was independent of protein abundance in erlotinib-resistant (Figure 3E) and cabozantinib-resistant (Figure 4E) cells. Within the quantified phosphoproteome of resistant cells versus sensitive cells, most phosphorylation sites were serine residues, followed by threonine and non-tyrosine sites (Figure 3F and Figure 4F).

### 3.5. Kinase Expression and/or Activity Is Elevated in Erlotinib- and Cabozantinib-Resistant TNBC Cells

Given the critical importance of kinases in regulating cancer cell behavior and response to treatment, we further focused on identifying kinases whose expression and/or activity were altered as part of acquisition to erlotinib or cabozantinib resistance in TNBC cells. From the LC-MS/MS analysis of erlotinib-resistant (468LR) versus erlotinib-sensitive (468S) cells, we quantified 106 unique kinases across the three datasets, with the majority quantified from the kinomics dataset (*n* = 92), and some unique kinases (*n* = 14) quantified in the proteomics and phosphoproteomic datasets (Figure 5A). From the analysis of cabozantinib-resistant (231LR) versus cabozantinib-sensitive (231S) cells, we quantified 84 unique kinases across the three datasets, with many (*n* = 65) being identified from the kinomics dataset, although some (*n* = 9) were uniquely from the proteomic and phosphoproteomic datasets (Figure 5A). When assessing the distribution of quantified kinases across the kinome families in both cabozantinib-resistant and erlotinib-resistant experiments, kinases were quantified from each kinase family except for the RGC family, with most quantified kinases from the CMGC, tyrosine-kinase, and atypical families (Figure 5B). Similar to our observations with the volcano plots of the proteome and kinome (Figure 4), very few proteins were significantly altered with the acquisition of erlotinib- or cabozantinib-resistance, and there were no significant changes in kinase expression in cabozantinib-resistant MDA-MB-231 cells relative to sensitive controls (Figure 5C). However, expression of the kinases SYK, RIOK2, MAP3K2 (mitogen-activated protein kinase kinase 2; MEKK2) and MAP3K3 (mitogen-activated protein kinase kinase 3; MEKK3) were significantly elevated in erlotinib-resistant MDA-MB-468 cells compared to sensitive controls (*p* < 0.05) (Figure 5D).

We next used the phosphoproteomic-exclusive method KSEA (kinase-substrate enrichment analysis) and integrated phosphoproteomic–kinomic method KSEAM (modified KSEA) for predicting kinase activity. These methods rely on the use of kinase-substrate interaction databases to predict and enrich for kinases responsible for regulating the changes in phosphorylation observed in the phosphoproteome, and in the KSEAM method, that score is weighted based on the change in expression measured from the kinome. In erlotinib-resistant 468LR cells, we identified AKT1 and RPS6KA2 (RSK3; ribosomal protein S6 kinase alpha-2) as having significantly elevated KSEA scores relative to sensitive controls (*p* < 0.05) (Figure 6A), as well as several other kinases with significantly elevated KSEAM scores, including CSNK2A1 (CK2A1; casein kinase II subunit alpha), CAMK2D (calcium/calmodulin-dependent protein kinase type II delta chain), PRKDC (DNA-dependent protein kinase, catalytic subunit), MAPK3 (ERK1; mitogen-activated protein kinase 3), MAP2K2 (MEK2; MAPK kinase 2), PAK2, PRKCD (Protein kinase C-delta), and ARAF (*p* < 0.05) (Figure 6B). PAK2 was the only kinase with significant elevated scores for both KSEA and KSEAM (*p* < 0.05). Interestingly, there were several kinases with statistically significant negative KSEAM scores, including LIMK2, MAPK13, NEK2, CDK5, RPS6KA1 (RSK1), RAF1, GSK3α, MAPK8 (JNK1), and CDK2 (*p* < 0.05), indicating that these kinases were downregulated in response to acquisition of erlotinib-resistance.

In cabozantinib-resistant 231LR cells, the kinases LIMK2 (LIM domain kinase 2), LIMK1 (LIM domain kinase 1), PRKG1 (cGMP-dependent protein kinase 1, alpha isozyme), IKBKB (IKK-β; inhibitor of nuclear factor kappa-B kinase subunit beta), CHUK (IKK-α; inhibitor of nuclear factor kappa-B kinase subunit alpha), and PRKD1 (serine/threonine-protein kinase D1) had significantly elevated KSEA scores (*p* < 0.05) (Figure 7A). Using KSEAM analysis, CDK1 was the only kinase with a significantly elevated KSEAM score (but not KSEA score) in response to the acquisition of cabozantinib resistance (*p* < 0.05), with an elevated but non-significant KSEA score. In contrast, the kinases CDK7 and CSNK2A1 (CK2A1; casein kinase II subunit alpha) had significantly elevated KSEA but not KSEAM scores (*p* < 0.05) (Figure 7B), suggesting that these kinases could be potential mediators of cabozantinib resistance.

### 3.6. Resistance to Erlotinib Can Be Reversed in MDA-MB-468 Cells via Synergistic Targeting of AKT Activity

As proof of principle that the information gained from our integrated phosphoproteomic and kinomics analysis could be applied for therapeutic benefit, we carried out functional experiments in MDA-MB-468 cells with a focus on inhibiting AKT activity using the AKT inhibitor VIII (AKTi-1/2) and ERK1/2 activity with Temuterkib. With ERK1/2 activity inhibition, we observed attenuated colony formation of erlotinib-resistant and erlotinib-sensitive models. We observed that treatment with increasing concentrations of AKT inhibitor VIII (0.1 nM–1 μM) or Temuterkib- (0.1 nM–1 μM) attenuated colony formation in both erlotinib-resistant (468LR, 468HR) and erlotinib-sensitive (468S) models compared to treatment with erlotinib alone (0.1 nM–1 μM), an effect that was more pronounced with increasing concentrations of erlotinib (*p* ≤ 0.05) (Figure 8A). In assessing the effects of erlotinib ± AKT inhibitor VIII on EGFR and AKT signaling, we did not observe any significant changes in EGFR or ERK1/2 phosphorylation among treatment groups. However, there was a significant reduction in phosphorylation of AKT (Ser473) in erlotinib-resistant cell models (468LR and 468HR) treated with AKT inhibitor VIII alone or in combination with erlotinib relative to sensitive controls (*p* ≤ 0.05) (Figure 8B, Appendix A). To determine whether erlotinib and AKT inhibitor VIII or Temuterkib are acting synergistically or independently in MDA-MB-468 TNBC cells, we used the Synergy Finder platform [64] to compute the synergy scores of the inhibitor combinations using the four current models of synergy: Bliss independence model; Loewe additivity model; HSA (Highest Single Agent Model), and ZIP (Zero Interaction Potency) model [68,69]. The score outputs are the differences between measured inhibition and expected inhibition based on the synergy model for each respective inhibitor combination and cell line, thus, strong synergy or strong antagonism is likely to occur if the synergy scores are all greater than 10 or lesser than -10, respectively. For the erlotinib and AKT inhibitor VIII combinations (Figure 8C, Appendix A), we observed likely strong synergy in the 468LR model, with all four synergy scores >10 (ZIP = 14.12; HSA = 10.04; Loewe = 10.59; Bliss = 13.04).

We also observed a likely weak synergy in the 468HR model; with all four synergy scores >5 (ZIP = 5.77; HSA = 9.74; Loewe = 10.06; Bliss = 5.10). Erlotinib and AKT inhibitor VIII appear to be non-interactive in the erlotinib-sensitive 468S model, with synergy scores more inconsistent and closer to expected values (ZIP = 0.86; HSA = 6.70; Loewe = 8.19; Bliss = −1.15). For the erlotinib and ERK inhibitor combination (Appendix A), we observed a potentially weak synergy in the 468HR model with two synergy scores >5 (ZIP = 3.76; HAS = 6.09; Loewe = 5.45; Bliss = 1.46) and observed likely independence in the 468LR model with all four synergy scores around 0 (ZIP = 1.00; HAS = 2.02; Loewe = 2.45; Bliss = 1.32).

## 4. Discussion

Over the last several decades, tremendous progress has been made in the identification and development of new targeted therapies against cancer. One of the main challenges in maximizing patient benefit from these drugs is the development of therapeutic resistance, particularly for those therapies designed to target kinases and their associated downstream pathways [42,52,70,71]. Elucidation of the mechanisms contributing to acquired resistance to kinase inhibitors is therefore crucial for the successful clinical management of patients treated with these agents. In the specific context of triple-negative breast cancer (TNBC), the search for potential therapeutic targets has led to the study of receptor tyrosine kinases (RTKs), such as epidermal growth factor receptor (EGFR) and hepatocyte growth factor receptor (c-Met), which are often overexpressed in TNBC tumors and associated with poor prognosis [14,17,72]. While both the EGFR inhibitor erlotinib and the c-Met inhibitor cabozantinib have demonstrated promising efficacy in controlling initial disease progression in clinical trials, in most cases, patients develop resistance to treatment [28,29,39,73].

In the current study, we hypothesized that the upregulation of kinases within the EGFR and c-Met signaling axes contributes to the development of acquired erlotinib and cabozantinib resistance in TNBC. We developed two separate models for erlotinib and cabozantinib resistance using the MDA-MB-468 and MDA-MB-231 human breast cancer cell lines, respectively. Acquired resistance models were established by applying long-term exposure to and selective pressure on cultured TNBC cells using a clinically relevant concentration of cabozantinib or erlotinib (defined as a low-resistance model) or long-term exposure and selective pressure using a step-wise escalation method to a supra-clinical concentration (defined as a high-resistance model). Comparing these resistant models to sensitive parental controls, we observed that erlotinib- or cabozantinib-resistant TNBC cells demonstrate enhanced cell proliferation, migration, invasion, and/or activation of canonical downstream signaling. Using a SILAC-labeled quantitative mass spectrometry approach, we assessed the effects of erlotinib or cabozantinib resistance on the phosphoproteome, proteome, and kinome via a KSEA/KSEAM integrated analysis approach to assess kinase activity changes. We identified an upregulation of AKT1, CK2A1, and ERK1 activity and SYK expression in erlotinib-resistant cells and an upregulation of CDK1, CDK7, and CK2A1 activity in cabozantinib-resistant cells. Functional inhibitor studies revealed inhibitor synergy between erlotinib and the AKT inhibitor in MDA-MB-468 TNBC cells, suggesting that AKT1 contributes to erlotinib resistance in our resistant MDA-MB-468 cell line model.

We opted for two different techniques to develop our resistant cell models to simulate a clinically relevant model (low-dose resistance) and a more robust resistant model (high-dose resistance). Unexpectedly, the low-dose resistant models for erlotinib and cabozantinib had higher-fold resistance factors than the high-dose resistant models. This may be due to the time duration of the selective pressure period, as the high-dose model was developed using a dose-escalation method, and thus those cells were under high-dose selection for a shorter period (3–4 sub-passages) than the low-dose cells were under their consistent low-dose (10 sub-passages). For future studies, this could be mitigated by dose escalating sooner and/or starting the development of low-dose resistant models once the high-dose models have reached their high dose. We also observed that erlotinib-resistant MDA-MB-468 cells had higher resistance than the cabozantinib-resistant MDA-MB-231 cells. This may have been due to the parental MDA-MB-231 cells being less sensitive to cabozantinib than the parental MDA-MB-468 cells were sensitive to erlotinib; thus, it may have required higher concentrations of cabozantinib to induce a more significant increase in resistance. However, there was still a >2× increase in the IC_50_ cell proliferation values of cabozantinib in resistant cells compared to sensitive, and recovery of AKT1 signaling in resistant cells treated with cabozantinib, which provided sufficient rationale to continue investigations with the cabozantinib-resistant MDA-MB-231 model.

Kinase activity and expression analysis from the mass spectrometry analysis proteomics demonstrated that the MAPK signaling pathway is heavily involved in the acquired erlotinib resistance in our MDA-MB-468 cell model. We observed increased activity of the kinases in the ERK cascade, including A-raf, MEK2 (MAPK2), and ERK1 (MAPK3), as well as the kinase PAK2, which is involved in the JNK signaling cascade. We also observed increased expression of the kinases MEKK2 (MAP3K2) and MEKK3 (MAP3K3), which are involved with the MAPK5 signaling cascade based on pathways found in the KEGG database [74,75,76]. As shown in a previous work [55], treatment with erlotinib results in attenuation of ERK1/2 signaling in erlotinib-sensitive TNBC cells as it is a significant downstream regulator of EGFR signaling, particularly with regard to cell proliferation and migration [55]. Therefore, increased activity and signaling of MAPK pathway kinases could counteract the effects of erlotinib as resistance is acquired, potentially compensating for the loss of EGFR signaling. It is interesting that in the synergistic assay between Temuterkib and erlotinib, we only observed a potentially weak synergistic or independent relationship between the two inhibitors in our resistant models. However, the resistant models were still responsive to ERK inhibition alone without the presence of erlotinib, thus still presenting as an option for targeting in situations of erlotinib resistance.

The kinases RIOK2 and SYK were also significantly upregulated in erlotinib-resistant MDA-MB-468 TNBC cells but did not demonstrate a change in activity in the KSEA/KSEAM analysis. RIOK2 is an atypical kinase involved in the maturation of the 40S ribosomal subunit of the ribosome and thus may play a role in protein expression regulation [77] but is unlikely to play a role in erlotinib resistance. Unlike RIOK2, there is a biological rationale for the non-receptor tyrosine kinase SYK to potentially contribute to erlotinib resistance. SYK positively regulates EGFR and has been shown to be involved in EGFR signaling in squamous cells and ovarian carcinoma, where it contributes to paclitaxel and lapatinib resistance [78,79]. However, it has been reported that SYK possesses both tumor promoter and suppressor roles in human breast carcinoma and that decreased expression is associated with a poor prognosis [80,81,82]; thus, SYK inhibition may work either synergistically or antagonistically with erlotinib.

Given the role of the PI3K/AKT signaling pathway in cell proliferation and survival, the increased activity of AKT1 from the KSEA analysis of the low-resistance model and increased phosphorylation of AKT1 in the immunoblots of the high-resistance model suggested that AKT1 was likely contributing to the erlotinib resistance. The likely synergy observed between AKT inhibition and erlotinib in our erlotinib-resistant models, compared to the non-interaction observed in the sensitive model, further confirms the contribution of AKT signaling to erlotinib resistance. If AKT was not contributing to erlotinib resistance, we would have expected to see a non-interactive or potentially antagonistic relationship between erlotinib and AKT inhibitor VIII in the erlotinib-resistant cells. However, since AKT inhibitor VIII is a selective combination AKT1/AKT2 inhibitor, we were not able to elucidate the relatively distinct contributions of AKT1 versus AKT2 in the inhibitor synergy assays. Thus, future studies could use selective AKT1 (e.g., A-674563) [83] and AKT2 (e.g., CCT128930) [84] inhibitors in combination synergy assays with erlotinib. These assays could also include a combination of AKT inhibitors with other EGFR inhibitors, such as lapatinib or gefitinib, as well as an investigation of the potential compensatory role of ERK/MEK and SYK in rescuing erlotinib resistance in TNBC. Finally, expression and/or activity of AKT, ERK/MEK, and/or SYK could be investigated as potential prognostic factors in predicting sensitivity/resistance to EGFR inhibitor treatment in TNBC patients as is being undertaken in NSCLC [85,86,87].

From the integrated KSEA and KSEAM analysis of cabozantinib-resistant MDA-MB-231 cells, we identified several kinases that were upregulated with acquired cabozantinib resistance, with CK2 (casein kinase II), CDK1, and CDK7 presenting as the most promising potential targets. CK2 is an important regulator in inhibiting apoptosis and promoting DNA damage repair in cancer and, thus, is a promising potential target for cancer therapeutics [88]. With the role that CDK1 and CDK7 play in the regulation of cell-cycle checkpoints, these two kinases have previously been shown to be useful targets for the inhibition of cell proliferation of TNBC cells [89,90]. Based on the KSEA/KSEAM analysis, other potential kinases that may be involved include IKK kinases, which may be involved in tumor development via NF-kB pathway regulation [91], and LIMK, which are LIM kinases that are heavily involved in the regulation of cell migration through actin filament regulation [92]. Future investigations could include functional inhibitor studies to validate the potential mediators of resistance identified in the kinase activity analysis, including CK2A (e.g., CX-4945) [88], CDK1 (e.g., RO-3306) [93], and CDK7 (e.g., CT7001 or SY-1365) [94].

The poor coverage of phospho-tyrosine phosphopeptides using the TiO_2_ enrichment technique limits the ability to assess changes in the kinase activity of tyrosine kinases such as SYK. Thus, expanding phosphopeptide sample enrichment with deep phospho-tyrosine enrichment would expand the coverage of kinases evaluated in KSEA/KSEAM. Similarly, expanding kinome coverage in the kinomics sample enrichment would expand the coverage of kinases evaluated in the KSEAM-specific portion of the analysis. It would have also been interesting to compare the 468LR and 468HR models and assess the treated cell lines against untreated controls using the integrated MS analysis to identify different mechanisms underlying the two resistance models and delineate the effects of the inhibitors on global proteome levels. Finally, one of the ways cells acquire resistance to EGFR inhibitors is through mutation in EGFR itself. Future studies are therefore aimed at sequencing the EGFR gene in 468LR and 468 HR to assess gatekeeper mutations such as T790M or others [95].

## 5. Conclusions

In conclusion, the results of the current study support the concept that information gained from integrated phosphoproteomic and kinomics analysis may be useful for the purposes of enhancing therapeutic benefit, particularly in the setting of acquired resistance to treatment with erlotinib and cabozantinib in TNBC. Using this integrated proteomics approach, we identified several potential kinase mediators of cabozantinib-resistance and confirmed the contribution of AKT1 to erlotinib-resistance in TNBC-resistant cell lines, with potential involvement from the MEK/ERK signaling pathway, which could present as potential targets in combinatorial therapy to overcome clinical erlotinib resistance. Finally, our work suggests a potential role for CK2, CDK1, and CDK7 in cabozantinib-resistance of TNBC cells. The resistance mediators identified in this study could be applied in the future to develop new therapeutic strategies for counteracting resistance to EGFR and c-Met inhibitors in triple-negative breast cancer.

## Figures and Tables

**Figure 1 biomedicines-11-02406-f001:**
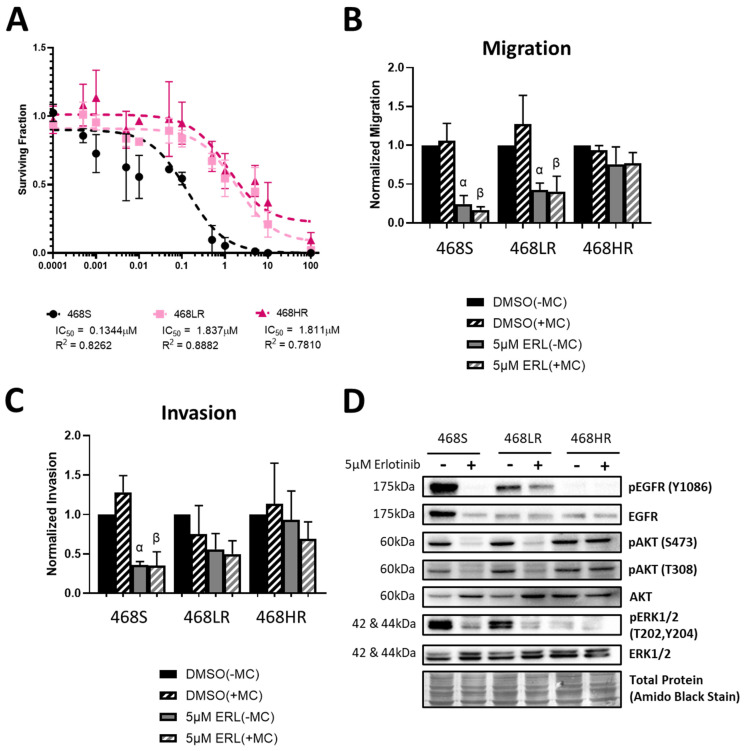
Erlotinib-resistant human MDA-MB-468 TNBC cells demonstrate a lack of inhibition of cell proliferation, migration, invasion, and EGFR/ERK signaling. MDA-MB-468 cells with low-resistance (468LR) or high-resistance (468HR) to erlotinib were developed as outlined in the Materials and Methods and Appendix A. (**A**) Cells were treated with erlotinib (ERL; 0.1 nM–100 μM) or DMSO (vehicle control) and subjected to colony-forming assays over 10–14 days, with media and inhibitors refreshed every 4 days. Non-linear regression was used to determine IC_50_ values. (**B**,**C**) MDA-MB-468 cell models were subjected to transwell migration (**B**) and invasion (**C**) assays (24 h) in the presence of erlotinib (5 µM) or DMSO vehicle control for 24 h, ±10 µg/mL Mitomycin C (MC; to control for proliferation). The number of migratory or invasive cells was normalized to 468S cells treated with DMSO (-Mitomycin C) (*n* = 3). α = significantly different than the respective DMSO (−MC) control. β = significantly different than the respective DMSO (+MC) control. (**D**) Representative immunoblots of cell lysates from 468S, 468LR, and 468HR cells treated with either 5 µM ERL or DMSO for 24 h, with Amido Black staining used for loading control. The corresponding quantitative densitometric analysis is presented in Appendix A.

**Figure 2 biomedicines-11-02406-f002:**
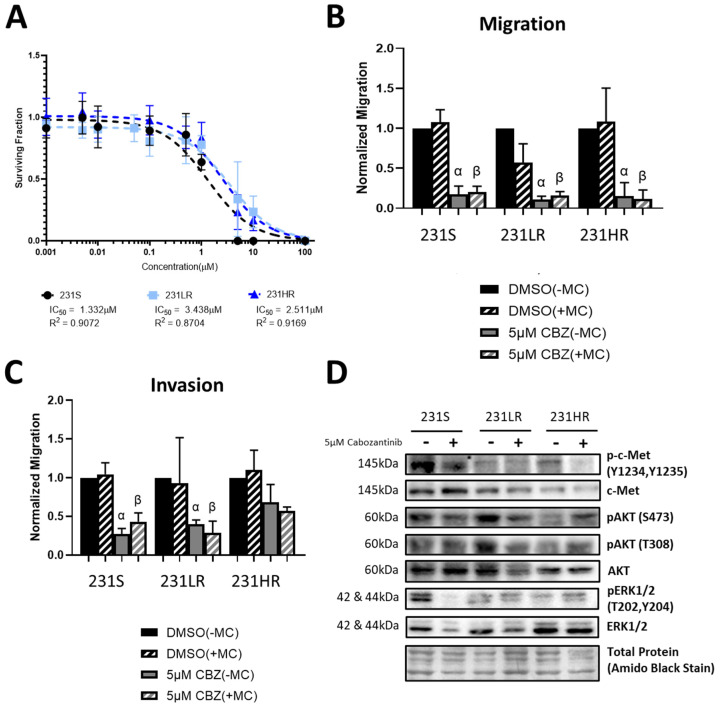
Cabozantinib-resistant human MDA-MB-431 TNBC cells demonstrate a lack of inhibition of cell proliferation, invasion, and AKT signaling. MDA-MB-231 cells with low-resistance (231LR) or high-resistance (231HR) to cabozantinib were developed as outlined in the Materials and Methods and Appendix A. (**A**) Cells were treated with cabozantinib (CBZ; 0.1 nM–100 μM) and subjected to colony-forming assays over 10–14 days, with media and inhibitors refreshed every 4 days. Non-linear regression was used to determine IC_50_ values. (**B**,**C**) MDA-MB-231 cell models were subjected to transwell migration (**B**) and invasion (**C**) assays (24 h) in the presence of cabozantinib (5 µM) or DMSO vehicle control for 24 h ± 10 µg/mL Mitomycin C (MC) to control for proliferation. The number of migratory or invasive cells was normalized to 231S treated with DMSO (-Mitomycin C) control (*n* = 3). α = significantly different than respective DMSO (−MC) control. β = significantly different than the respective DMSO (+MC) control. (**D**) Representative immunoblots of cell lysates from the resistant and sensitive cell lines treated with either 5 µM cabozantinib or DMSO control for 24 h, with Amido Black staining used for loading control. The corresponding quantitative densitometric analysis is presented in Appendix A.

**Figure 3 biomedicines-11-02406-f003:**
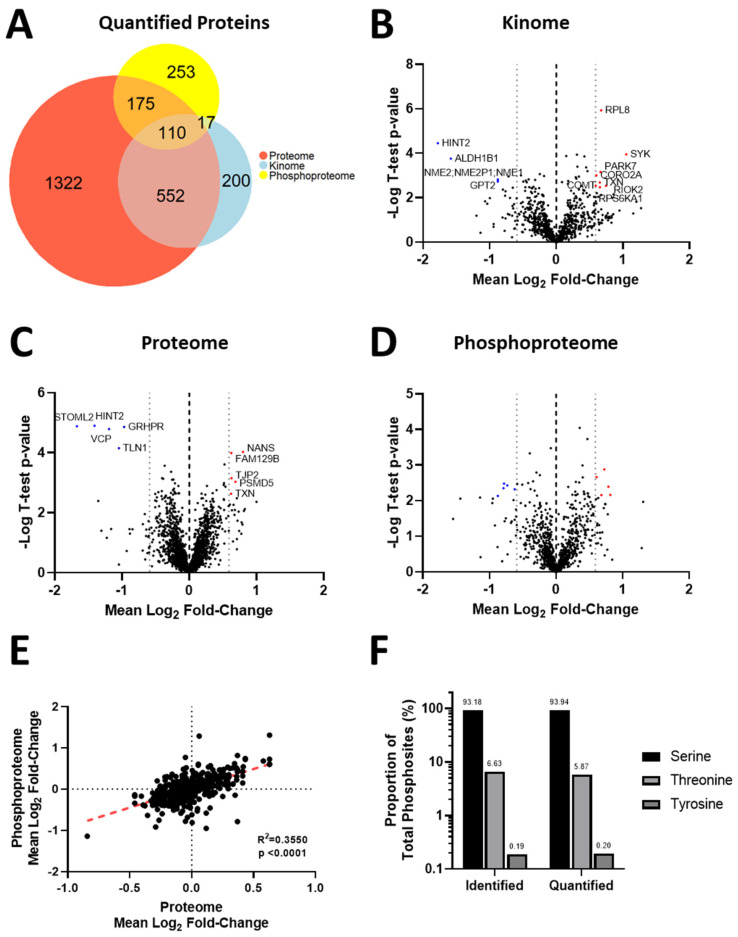
Acquisition of resistance to erlotinib alters the proteome and kinome of MDA-MB-468 TNBC cells with minimal change to the phosphoproteome. SILAC-labeled TNBC cells, including erlotinib-resistant (468LR) versus erlotinib-sensitive (468S) cells, were treated with erlotinib (5 µM) or DMSO vehicle control for 24 h before being subjected to LC-MS/MS, as described in Materials and Methods. (**A**) Euler diagrams of the overlap of proteins quantified from proteomic (*n* = 2159), kinomic (*n* = 879), and phosphoproteomic (number of proteins with quantified phosphosite = 555) with significant overlap among the three datasets. (**B**–**D**) Volcano plots for quantified proteins from the (**B**) kinome, (**C**) proteome, and (**D**) phosphoproteome showing significantly altered proteins or phosphosites in the 468LR versus 468S cell models. (**E**) The plot of phosphorylation sites versus protein abundance to assess whether protein abundance biases respective phosphorylation changes. (**F**) Quantification and distribution of phosphorylation sites identified.

**Figure 4 biomedicines-11-02406-f004:**
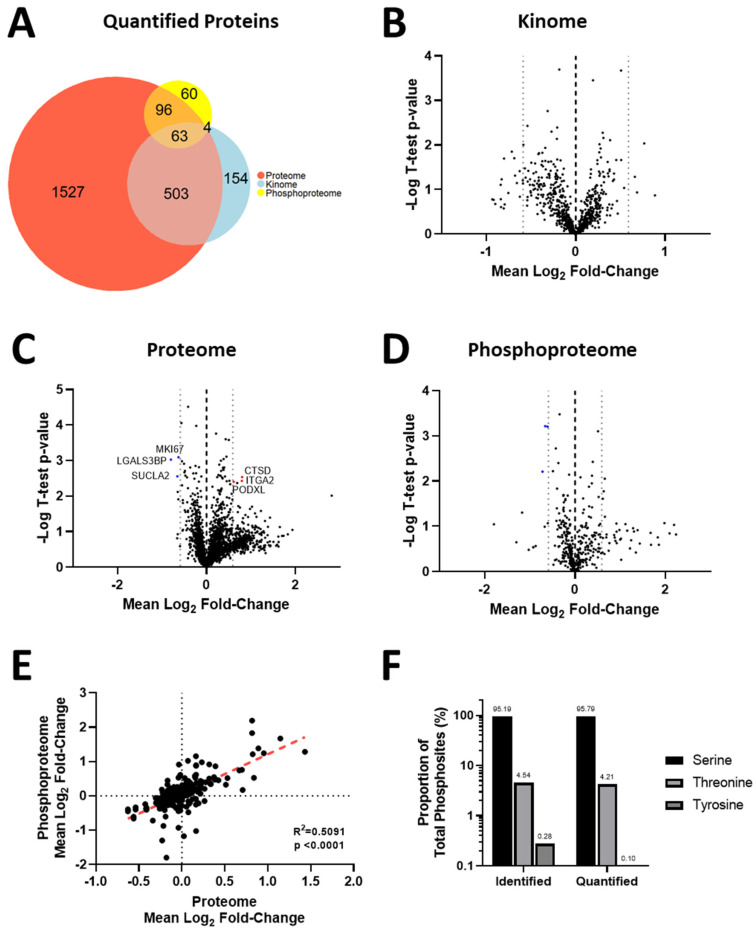
Acquisition of resistance to cabozantinib alters the proteome and kinome of MDA-MB-231 TNBC cells with minimal change to the phosphoproteome. SILAC-labeled TNBC cells, including cabozantinib-resistant (231LR) versus cabozantinib-sensitive (231S) cells, were treated with cabozantinib (5 µM) or DMSO vehicle control for 24 h before being subjected to LC-MS/MS, as described in Materials and Methods. (**A**) Euler diagrams of the overlap of proteins quantified from proteomic (*n* = 2189), kinomic (*n* = 724), and phosphoproteomic (number of proteins with quantified phosphosite = 223) with significant overlap among the three datasets. (**B**–**D**) Volcano plots for quantified proteins from the (**B**) kinome, (**C**) proteome, and (**D**) phosphoproteome showing significantly altered proteins or phosphosites in the 231LR versus 231S cell models. (**E**) The plot of phosphorylation sites versus protein abundance to assess whether protein abundance biases respective phosphorylation changes. (**F**) Quantification and distribution of phosphorylation sites identified.

**Figure 5 biomedicines-11-02406-f005:**
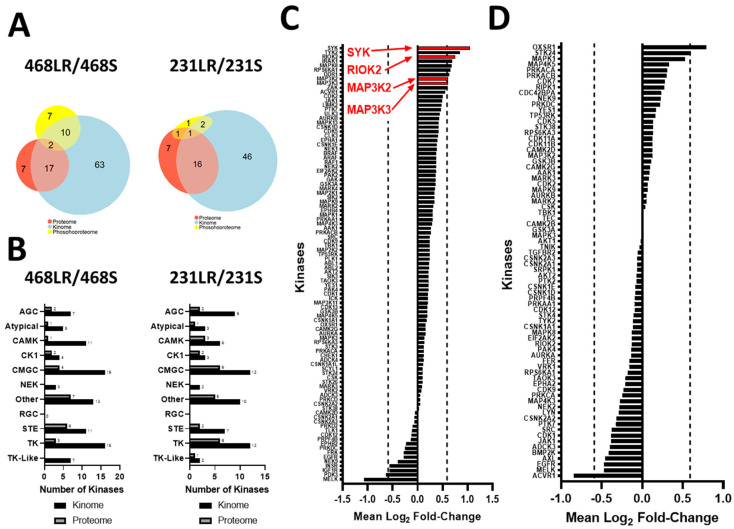
Kinase expression of SYK, RIOK2, MAP3K2, and MAP3K3 is elevated in erlotinib-resistant MDA-MB-468 TNBC. (**A**) Euler diagrams of the overlap of kinases quantified from erlotinib-resistant 486LR cells (*left panels*) or cabozantinib-resistant 231LR (*right panels*) compared to the respective sensitive controls. (**B**) Distribution of kinases from each of the kinase families quantified from erlotinib-resistant (468LR) versus sensitive (468S) (*left panels*) and cabozantinib-resistant (231LR) vs. sensitive (231S) (*right panels*) proteome and kinome. (**C**,**D**) Average logarithmic (base 2) fold-change for kinases quantified in kinomic and proteomic experiments of (**C**) 468LR/468S and (**D**) 231LR/231S. Dashed lines demarcate the threshold for biological significance (1.5×, ±0.59 in log2 base). Statistically significant increases in kinases are indicated in red.

**Figure 6 biomedicines-11-02406-f006:**
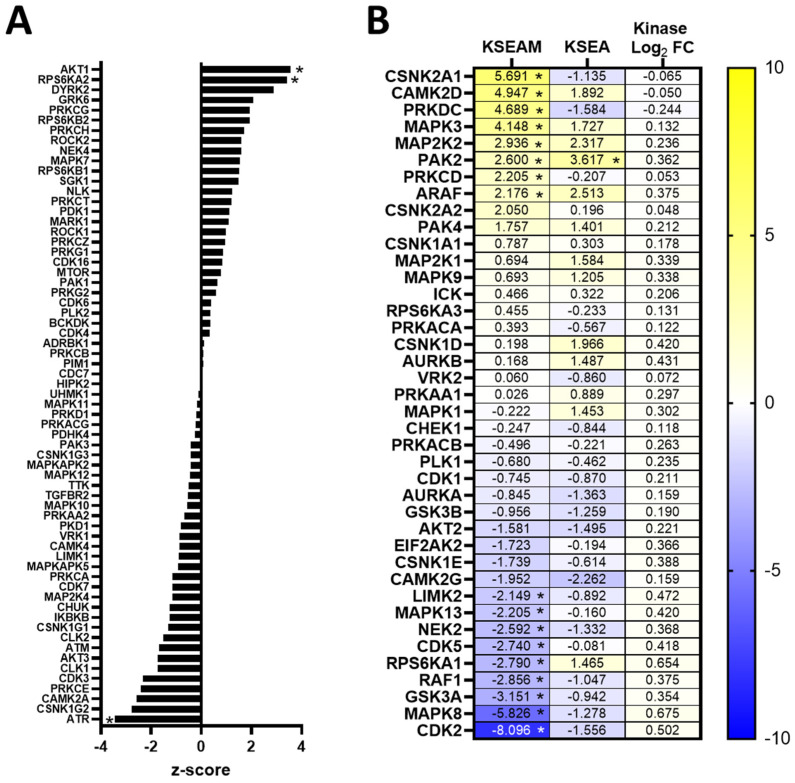
Kinase activity of AKT1, MAPK3, and PAK2 is enhanced in erlotinib-resistant MDA-MB-468 TNBC cells. (**A**) Kinases with only KSEA scores (at least 2 substrates) were plotted. (**B**) Heatmap of kinases with KSEA and KSEAM scores and average logarithmic (base 2) fold-change for kinases expression in resistant/sensitive. Statistics for z-scores and kinase fold-changes were calculated using one-tailed Student’s *t*-tests controlled by the Benjamini–Hochberg FDR (False-Discover Rate) procedure. * = corrected *p*-value of <0.05 at FDR = 0.1.

**Figure 7 biomedicines-11-02406-f007:**
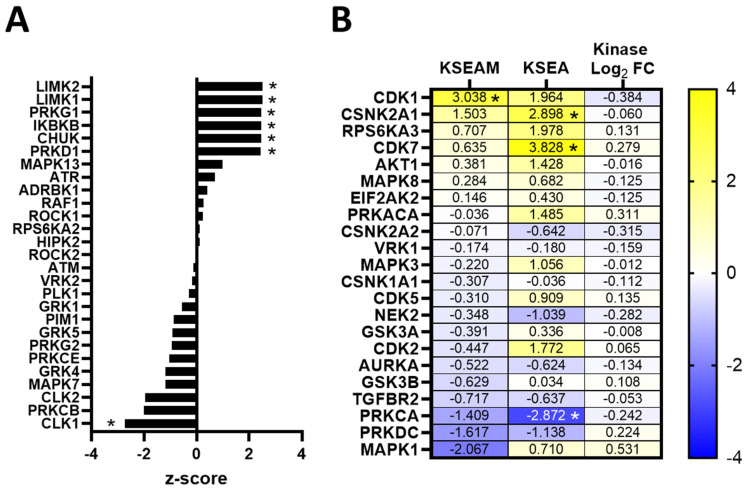
Kinase activity of CDK1, CDK7, and CSNK2A1 is elevated in cabozantinib-resistant MDA-MB-231 TNBC cells. (**A**) Kinases with only KSEA scores (at least 2 substrates) were plotted. (**B**) Heatmap of kinases with KSEA and KSEAM scores and average logarithmic (base 2) fold-change for kinase expression in resistant/sensitive. Statistics for z-scores and kinase fold-changes were calculated by a one-tailed Student’s *t*-test controlled by the Benjamini–Hochberg FDR (False-Discover Rate) procedure. * = corrected *p*-value < 0.05 at FDR = 0.1.

**Figure 8 biomedicines-11-02406-f008:**
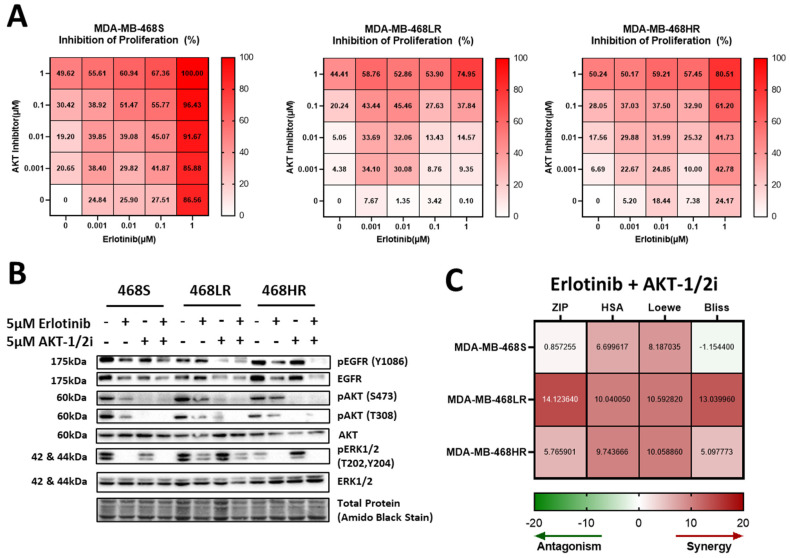
Resistance to erlotinib can be rescued in MDA-MB-468 cells via the synergistic targeting of AKT activity. MDA-MB-468 cells were treated with inhibitors and subjected to colony-forming assays over 10–14 days, with media and inhibitors refreshed every 4 days. (**A**) Heatmaps of percent inhibition for the erlotinib + AKT inhibitor VIII combination treatments of the sensitive- (468S; *left panel*) and resistant- (468LR; *middle panel* and 468HR; *right panel*) cell lines. Inhibition was determined by measuring the surviving fraction of colony-forming assays with a matrix of concentrations (0 µM, 0.001 µM, 0.01 µM, 0.1 µM, and 1 µM) for each inhibitor in each inhibitor combination. (**B**) Representative immunoblots of cell lysates from resistant- and sensitive-MDA-MB-468 cells treated with either 5 µM erlotinib and/or 5 µM AKT VIII inhibitor (AKT-1/2i) or with DMSO control. The corresponding quantitative densitometric analysis is presented in Appendix A. (**C**) Heatmaps of the synergy scores (ZIP, HSA, Loewe, and Bliss) of the inhibitor combinations erlotinib + AKT VIII inhibitor treatment for the resistant and sensitive cell lines.

## Data Availability

The mass spectrometry data used for analysis in this manuscript is accessible as public datasets and can be found here: https://repository.jpostdb.org/preview/421236292649c4bcc1715d (accessed on 26 June 2023).

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
