# Peer review of "Involvement of the AKT Pathway in Resistance to Erlotinib and Cabozantinib in Triple-Negative Breast Cancer Cell Lines"

_biomedicines, 2023, doi:10.3390/biomedicines11092406_

Round 1

Reviewer 1 Report

Lefebvre et al. describe the effect of TKI resistance (cabozantinib and erlotinib) in TNBC models (MDA-MB-231 and MDA-MB-468). The manuscript is well designed, well written, easy to read. However, I have some questions:

  1. The authors report migration assays in gelatin-coated Boyden chambers, however, this experimental strategy is not common. Migration assays are generally performed only with uncoated Boyden chambers. Why use gelatin (whose component is collagen) in these experiments?

  2. The time in which the cell lysates were obtained is not indicated, 12 h, or 24 h? Why analyze protein phosphorylation at a point in time? In these cases, it is recommended to analyze the signaling pathways through a time course assays, since post-translation modifications are fast and dynamic events.

Reviewer 2 Report

Summary and overall comments:

Resistance to tyrosine kinase inhibitors present a huge challenge in the clinic. Authors derived EGFR and MET inhibitor resistant cell lines through long-term exposure. Through phenotypic assays, molecular biology experiments and proteomics experiments, authors identified AKT pathway as a potential mechanism of adaptive resistance to EGFR inhibitor. This hypothesis was then tested with combination treatment of AKT and EGFR inhibitors. These results are important contribution in the field as they confirm several previous publications. However, there are major concerns and comments to address:

Major comments:

1.       Methods section lacks sufficient details about how the samples for proteomics were processed. For example, lysis buffer, type of desalting (C18, magnetic beads or others), trypsin to protein ratio, digestion conditions, etc. Additionally, details about kinome enrichment (protein incubation conditions, source of beads, etc) and phosphopeptide enrichment are missing. Either add the details (supplementary is fine) or cite appropriate paper(s).

2.       Labels for supplementary figures are incorrect. Please make sure figures are labeled appropriately and that they appear chronologically. There are two Figure S1 and two Figure S2.

3.       Please define how you calculated “T-test difference” in volcano plots in Figure 3 and 4. Is it log2 fold change between sensitive and resistant cell lines? If it is the T-value that’s plotted on x-axis, please explain why the T-value is not proportional to the P value in y-axis, since in T-tests they should be proportional i.e. higher the absolute T-value, lower the P-value?

4.       Line 378: “When assessing the distribution of quantified proteins from the erlotinib-resistant kinomics/proteomics (Figure 3B,C) and cabozantinib-resistant kinomics/proteomics (Figure 4B,C), the majority of proteins fell within the bounds of log2 mean fold-change of -0.59 and 0.59 (threshold for biological significant fold-change), indicating that the observed acquisition of RTK resistance significantly alters global protein expression. In contrast, analysis of the resistant versus sensitive cell models indicates minimal disruption in the phosphoproteome as a result of acquisition of erlotinib (Figure 3D) or cabozantinib (Figure 4D) resistance.”

a.       Majority of the proteins or phosphosites are within the bounds of log2 mean fold-change of -0.59 and 0.59 i.e. most of the proteins are not changing and thus RTK resistance does not significantly alter global protein expression. This does not support the statement, “ indicating that the observed acquisition of RTK resistance significantly alters global protein expression.” However, there are few selected proteins that are significantly changing.  

5.       Line 385: “Average fold-change expression of phosphorylation sites was plotted against the corresponding protein ratio to confirm that the change in the phosphoproteome was independent of protein abundance in erlotinib-resistant (Figure 3E) and cabozantinib-resistant (Figure 4E) cells.”

a.       Looking at the proteome vs phosphoproteome dot-plots (3E and 4E), at least to my eyes, they appear to have positive correlation. Please calculate Pearson’s correlation coefficient and associated P-value to support the statement.

6.       Given that there were minimal changes in phosphoproteome overall (Figure 3D, 4D), it is surprising to see a high number of kinases enriched in KSEA analysis (Figure 6 and 7). Could you please explain this discrepancy? Results from KSEA may have been over-interpreted. For KSEA analysis, only the significantly changing phosphosites (with Fold Change 1.5, P<0.05) should be used as input. Additionally, only the significantly enriched kinases should be plotted. Please plot the phosphorylation levels of tentative substrates of AKT1, RPS6KA2, CDK1 and CDK2, to check whether the substrates indeed have higher phosphorylation levels.

7.       One of the ways cells acquire resistance to EGFR inhibitors is through mutation in EGFR itself. Have authors sequenced EGFR gene in 468LR and 468 HR to check for gatekeeper mutations such as T790M or other mutations in EGFR?

8.       Authors found that activation of AKT is implicated in resistance to EGFR inhibitors. While this is not a novel finding, since it has been shown previously in several studies (https://jhoonline.biomedcentral.com/articles/10.1186/s13045-019-0759-9, https://molecular-cancer.biomedcentral.com/articles/10.1186/s12943-018-0793-1 ), one of the classical activation of PI3K/AKT pathway is through MET amplification/overexpression. Check for MET overexpression in 468LR/HR cell lines compared to 468S cells. This will strengthen the findings in the paper.

9.       Biological variation is understandable; however, non-reproducibility in some of the main targets shown in western blots is concerning. Comparing DMSO only conditions in Figure 1D and 8B, EGFR protein levels are lower in 468LR and 468HR compared to 468S in Figure 1D; however, EGFR levels are slightly higher in 468HR compared to 468S in Figure 8B. Another example is phosho-AKT levels: phospho-AKT is upregulated in response to erlotinib in 468HR cells in Figure 1D; however, phospho-AKT levels are downregulated in 468HR cells in Figure 8B. Given that these treatment conditions are same, they are essentially biological replicates. The observation of increased phospho-AKT levels is highlighted in the results sections as it leads to follow-up experiments with AKT inhibitor. The aforementioned non-reproducibility is concerning.

1.   For SILAC based proteomics experiments, authors compared erlotinib treated 468S and 468LR cells. These cells were treated with 5uM erlotinib and samples were acquired at 24-hour timepoint. In this case, authors are looking at differences in erlotinib induced changes in proteome/phosphoproteome between the 468S and 468LR cells. One of the better comparisons will be differences in baseline proteome/phosphoproteome between the two cell lines i.e. DMSO treated 468S and 468LR cells. From Figure 1D, it looks like 468LR cells have lower EGFR protein levels compared to 468S cells at the baseline levels (i.e. DMSO only treatment) suggesting that downregulation of EGFR protein levels may be a potential mechanism of resistance. Erlotinib treatment further leads to lower protein levels in 468S cells. Proteomics experiment comparing baseline proteome may help elucidate mechanisms of erlotinib resistance.  

1.   Additionally, minimal proteomic changes with 5uM erlotinib at 24-hour timepoint is surprising. Such high concentration of erlotinib would lead to cell death, >95% in 468S and >60% in 468LR (Based on Figure 1A), and will likely lead to global protein level changes. Could authors comment on this discrepancy?

Minor comments:

1.       To develop resistant cell lines, can you mention how many total days the cells were treated for? Or how often the cells were passaged?

2.       Discussion of figures is not chronological. For example, Figure 4subparts are discussed before Figure 3D, E, F.

3.       Figure 8A: For 468S heatmap, concentration of AKT inhibitor should be 0 instead of 0.0001 uM?

Reviewer 3 Report

In the current study, authors developed two separate models for cabozantinib and erlotinib resistance using the MDA-MB-231 and MDA-MB-468 cell lines, respectively. They observed that erlotinib- or cabozantinib-resistant cell lines demonstrate enhanced cell proliferation, migration, invasion and activation of EGFR or c-Met downstream signaling (respectively). With a SILAC (Stable Isotope Labeling of Amino acids in Cell Culture)-labeled quantitative mass spectrometry proteomics approach, they assessed the effects of erlotinib- or cabozantinib resistance on the phosphoproteome, proteome and kinome. Using this integrated proteomics approach, they identified several potential kinase mediators of cabozantinib-resistance and confirmed the contribution of AKT1 to erlotinib-resistance in TNBC resistant cell lines. This manuscript is comprehensive and deserves to be published after minor revision.

Here are the points:

“alters” should be “alter” in Line 347.

It is not clear that erlotinib is also effective on c-met.

It is also not clear that the use of erlotinib and cabozantinib is appropriate.

In Supplementary there is written “error” in page 8.

Minor editing of English language required.

Round 2

Reviewer 1 Report

The authors responded in detail to the doubts raised. However, I still have doubts about the usefulness of gelatin in migration assays in Boyden chambers, since the size of the pores in the membrane prevents the passage of cells due to reasons such as gravity.

Reviewer 2 Report

Authors have addressed comments and concerns.